# Anxiety levels among school-going adolescents in peri-urban areas of Karachi, Pakistan

Tooba Seemi[1], Hina Sharif[2]*, Sana Sharif[3], Hira Naeem[1], Farhat ul Ain Naeem[4], Zoya Fatima[4]

1 Research Associates SINA Health and Welfare Education Trust, University of Karachi, Karachi, Pakistan,
2 Assistant Manager Pharmacy and Research, SINA Health and Education Welfare Trust, Karachi, Pakistan,
3 School of Public Health, University of Saskatchewan, Saskatoon, Canada, 4 Aga Khan University, Karachi, Pakistan

* Hina.sharif@sina.pk

## Abstract

### Introduction

Mental health problems are pervasive nowadays. Adolescents are often expected to balance academic performance with familial obligations and work to support the family financially if they belong to low-socio-economic areas. These pressures can lead to Anxiety, stress, and even depression.

### Objective

The study's main objective is to assess the association of gender, parenting style, eating habit, and screen timing with the level of Anxiety among school-going adolescents in three peri-urban areas of Karachi, Pakistan.

### Methodology

A school-based cross-sectional study was conducted among adolescents of age group 10–19 years using the Generalized Anxiety Disorder-7 (GAD-7) scale in two months in, three populated peri-urban areas of Karachi, Pakistan. Collected Data was analyzed by statistical software SPSS version with 80% response rate.

### Finding

In our collected data, 544 students participated; 313 (57%) were female, and 231(42%) were male. Among them 173(33%) participants showed moderate to severe level of anxiety out of which 140(26%) were female and 33(7%) were male. As per our study, strict Parenting style plays a significant role in developing moderate to severe 108(20%) levels of Anxiety, shown among adolescents in the studied population. In addition to this, those who do not take tuition 115 (21%), do not use transport to school 91(16%), spend significant time in games on mobile and computer 101 (18%), and have no involvement in physical activities 172 (31%) show more moderate to severe level anxiety than others factors. A significant multivariate

**Data Availability Statement:** When it comes to sharing de-identified data sets, there are ethical and legal restrictions in place to protect individuals' privacy and prevent potential risks of re-identification. As Data contains potentially

identifying or sensitive patient information, it must be handled with utmost care to safeguard confidentiality. Such restrictions have been imposed by Research Ethics Committees to ensure that data sharing aligns with established ethical guidelines and complies with relevant data protection laws. These measures are put in place to maintain the privacy and confidentiality of the individuals involved in the research. our ERB committee email address is research.office@sina.pk.

**Funding:** The author(s) received no specific funding for this work.

**Competing interests:** The authors have declared that no competing interests exist.

**Abbreviations:** AOR, adjusted odd ratio; CI, Confidence Interval; COR, crude odds ratio; GAD-7, Generalized anxiety disorder-7 Scale; HAM-A, Hamilton Anxiety Rating Scale; HDRS, Hamilton Depression Rating Scale; p-value, probability value; SINA-ERB, Sina ethical review board; SPSS, Statistical Package for Social Science.

association between level of anxiety with gender, school commute, type of lunch Intake, smoker family member at home, physical activity, video game, tuition and strict parent.

## Conclusion

This study concluded that there are various factors which have great association with anxiety and can affect adolescents' mental health badly. The factors were parental strictness, video game playing, a sedentary lifestyle, and the smoking habits of family members. Children and adolescents must be evaluated as soon as possible while they are still young to prevent mental health issues.

## Introduction

*Anxiety* is a highly intense and considerable mental health problem affecting many populations, especially adolescents [1]. Globally, 6.5% of adolescents are estimated to have anxiety disorders [2]. Depression and Anxiety in adolescents is a psychological and emotional disease [3]. An effective screening tool should be brief, simple to administer and score, interpretable without significant or specialized training, practicable, and, if possible, free to use [4].

Different countries conducted studies on Anxiety in children using various tools to assess the level. The study showed good psychometric qualities in a recent study using adolescent survey data [5]. In addition, one of the recent studies conducted in the USA on young individuals (14–26) with substance use disorders through the GAD-7 scale also showed excellent internal consistency and constructed rationality by a cut score of 6 [6]. A European study shows Moderate to severe Anxiety in females (OR = 1.887, 95% CI: 1.446, 2.464), which concluded the reliability of the GAD-7 scale [7]. A study conducted in India shows 20% anxiety level in adolescents by using GAD-7 Scale [8]. Some researchers from the psychological side surveyed adults in Tertiary Health Care Centre, and they found that the reliable source of screening tools is GAD-7 and PHQ-9 [9]. In Pakistan, anxiety screening recently started for Professional adults, adolescents, and mothers; after the COVID-19 outbreak, several people suffered from different mental health problems [10, 11]. Still, there may be no studies that report adolescents' mental health.

In Pakistan, there is no concept of mental health screenings for school-going adolescents in a peri-urban area of Karachi, Pakistan. The growing age among children needs intense attention to grow healthy physically and mentally in their lives. Adolescents are often expected to balance academic performance with familial obligations, including contributing to household chores, taking care of younger siblings, and working to support the family financially. These pressures can lead to Anxiety, stress, and even adolescent burnout. Additionally, the education system in these areas is often characterized by low-quality teaching, overcrowded classrooms, and inadequate resources, all of which can lead to Anxiety and stress among students. Also, these areas have limited economic and financial resources due to illiteracy. People have no or less knowledge about mental health. Parents having no good relationship with their children also plays a great part in adolescents' Anxiety. Staying at home, with no family or friends gathering due to Covid-19, also devastates adolescents' mental health. As per our searched data, there are no reporting numbers about the anxiety assessment in school-going adolescents in peri-urban areas of Karachi, Pakistan. So, this is our study rationale. The goal is to assess the Anxiety in school-going adolescents of the highly vulnerable population and associate it with their demographics.

The study aims to assess and screen Anxiety among school-going adolescents in three populated peri-urban areas of Karachi, Pakistan, using the GAD-7 scale. The objectives of this study mainly surround the following points:

1. To assess the gender-based level of Anxiety among school-going adolescents in peri-urban areas of Karachi, Pakistan.

2. To determine the association between eating habits and anxiety levels among school-going adolescents in Karachi, Pakistan, and peri-urban areas.

3. To assess the association of parenting style and anxiety level among Karachi, Pakistan, school-going adolescents.

4. To assess the impact of screen time on the anxiety level of adolescents of peri-urban communities in Karachi, Pakistan.

## Material & method

### Study design

Cross-sectional study.

### Study participants

Adolescents aged 10–19 years were included in the study.

### Study site

This school-based cross-sectional study was conducted at different schools in three populated slums of Karachi, Pakistan, where people belong to a highly lower socioeconomic status. Karachi has around 18 slum locations, of which 40%, i.e., 82.83 million, live in urban areas [12] of the population. Out of these 18 towns [13], we selected three locations of highly populated slum areas on a random number table where people from different ethnic background resides, as shown in Fig 1. The selected regions were Orangi Town, North Karachi, and Gadap Town.

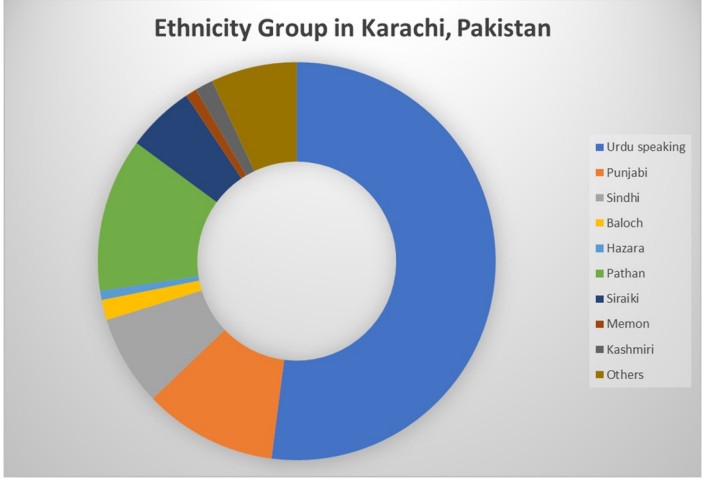

**Fig 1. Different ethnic groups in Karachi, Pakistan.**

### Ethics approval of the study and consent from children

The participants recruited after getting written consent from their parents were reported as unidentified. The first proposal was presented to the SINA Ethical Review Board (SINA-ERB) before the beginning of the study, which granted its approval via ERB no ERB0000011/09-22.

### Data size and duration

Since anxiety disorder prevalence is not known in the local population, especially in the peri-urban area, we assumed the prevalence of 50%, with a 95% confidence level with 5% absolute precision and a 20% no response rate inflated to 544 to be on the safe side.

### Data size calculation

$$\text{Participant's sample size} = (4pq/d2)*20\% \text{ NRR(no response rate)}$$

where p = anticipated the prevalence, q = (1-p), d = margin of error), d = 5%

$$\text{Participant's sample size} = ((4*50*(1-50))/(5*5))*20\%$$

$$\text{Participant's sample size} = 392*20\% = 470 + 23 = 493 + 51 = 544$$

### Inclusion and exclusion criteria

Children and adolescents between 10 and 19 years of age belong to the highly low-socio-economical areas of Karachi, Pakistan. They were admitted to schools in their locality and agreed to participate in the study. The parent's written consent was obtained before the commencement of the study. In contrast, those suffering from any identified psychological illness or any other disease or those who were reluctant to answer, omitted any question or did not understand due to a language barrier were excluded from the study.

### Data collection tools

The questionnaire consists of two sections:

The first is the student's consent section regarding whether they want to participate in the study.

### Demographic

A self-designed structured pre-tested questionnaire with details about the participants like name, age, gender, father's name, occupation, Ethnicity, educational information (taking tuition or not), parents' strictness, physical activities, and social activities.

## Generalized Anxiety Disorder Questionnaire-7 (GAD-Q-7)

The GAD-7 is a 7-item scale validated previously. The starting scores range from 0 to 3. It inspects how frequently the patient has experienced seven anxiety signs over the last two weeks, with responses like "not at all," "a few days," "more than half the days," and "almost daily" scoring 0, 1, 2, and 3 respectively [14].

### Data collection instrument validation

The pilot survey on 15% of the sample size of adolescents was done to ensure the validity of the self-designed structured questionnaire and get the response.

### Data collection process

Participants were included on a random basis after visiting seven schools (Government and private). The questionnaire was first translated into the local language before administering to the targeted population. Research first helped the questionnaire face-to-face to each child and adolescent of a selected age range during lunch break. The size of the sample of the targeted population included is shown in Fig 2.

### Anthropometric measurements

Blood pressure (systolic—SBP and diastolic—DBP), based on the average of two measurements taken on each adolescent (one at the start of data collection and another 10 to 15 minutes later) [15]. If the difference in SBP and DBP was greater than 10 mmHg, a third measurement was taken, using the mean of the lowest blood pressure measurements and excluding the highest [16].

### Sampling procedure

In order to guarantee the study's professionalism and scientific validity, a systematic random sampling technique was used. The stages in the procedure were as follows:

**i. School selection.** The School of Education & Literacy Department, Sindh, provided a comprehensive list of public schools situated in the slums of Orangi Town, North Karachi, Gadap town. Seven schools were chosen using the Probability Proportional to Size (PPS) technique from this list. This made sure that the schools picked were typical of the community.

**ii. Consent for the study.** The headmistress and teachers of the chosen schools gave their approval before any evaluations were carried out. The attendance book for each lesson was

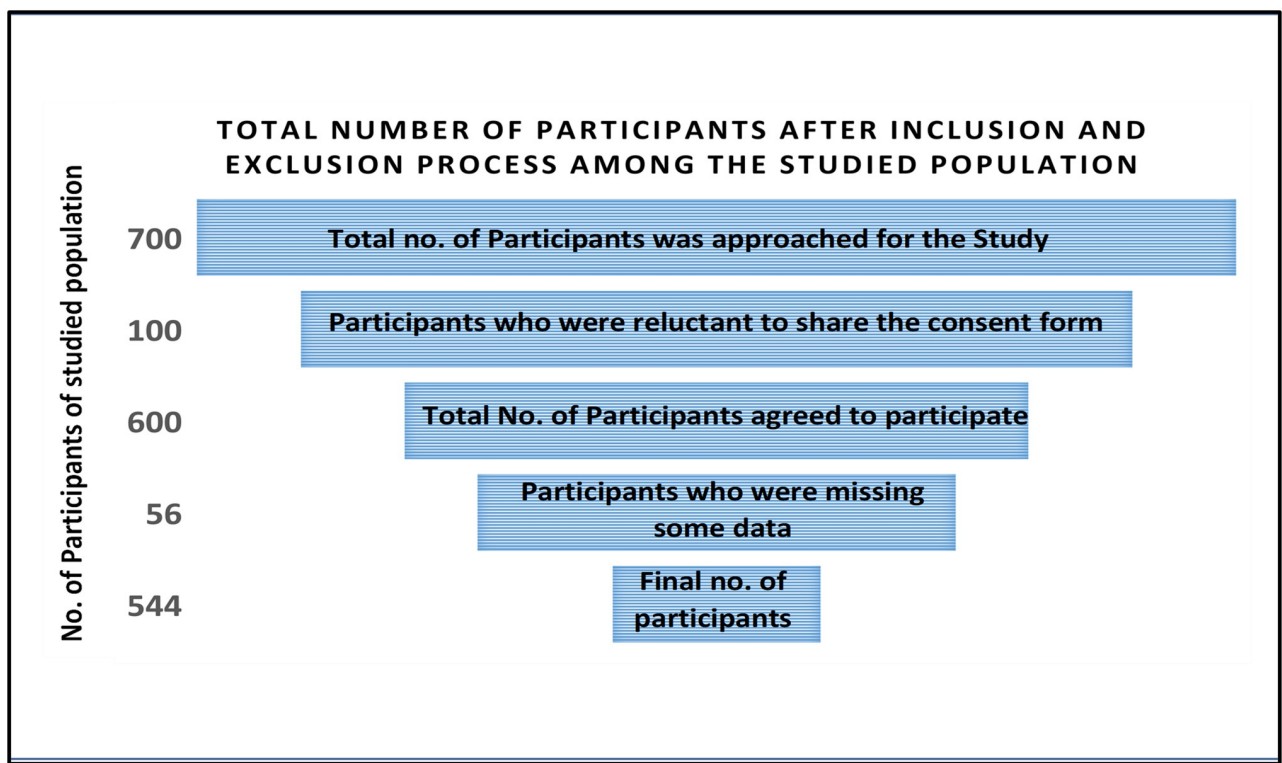

**Fig 2. Total number of participants after inclusion and exclusion process among the studied population.**

then used to enrol the students. Parents' written informed permission was collected, as well as the children's own agreement. Each student was given a personal identification number in order to ensure secrecy and anonymity.

 **iii. Collecting information.** Trained research colleagues took physical measures and delivered questionnaires to the enrolled students.

## Statistical analysis

All data obtained were reviewed for completeness and consistency before input into Microsoft Excel 2007 and exported to SPSS version 24 for analysis. The data were summarized through descriptive statistics (frequencies, tables, percentages, and averages) to describe the socio-demographic features, clinical variables, and Anxiety. The bivariate analysis was used to select the significant predictors. To avoid possible confounders, variables with p-values less than 0.05 in the bivariate model were added for further multivariate logistics regression. Variables with P-values less than 0.05 were deemed statistical predictors of the level of Anxiety in the multi-variate logistic model. The strength of the link was measured using the odds ratio with a 95% confidence interval.

## Result

A total of 700 participants were approached for the study with a 20% non-response rate, so 544 participants fulfilling the inclusion criteria were finally taken up for final analysis. Table 1, from a supplementary file, shows the socio-demographic analysis of the studied participants. There were 313 (57%) were female, and 231(42%) were male in total. The mean age of participants was 15 years (SD ±3). Maximum participants were female (57%), in the age group of 13–15 years (64%), majority of the participants belong to Urdu speaking 392(72%), consume junk food (90%), having a strict father (65%), after their school time, have no involvement in any physical activity (64%), spend significant time in playing video games (62%), not taking tuitions 319(59), majority 295(54%) having 4–6 members in family, have anyone is in family involve in smoking habit (86%). For the anxiety point of view, the females have severe to moderate level of anxiety (26%). There were 109 (20%) participants with the age range 13–15 years have moderate to severe level of anxiety. The 19% Urdu speaker study subjects had moderate to severe level of anxiety. The 153 (28%) participants consume canteen food had moderate to severe level of anxiety. Among the study participants with 4 to 6 sizes of household, there were 88 (16%) subjects had moderate to severe level of anxiety. There were 108 (20%) study subjects with strict father at home and who had a moderate to severe level of anxiety observed. There were 91 (16%) study participants who do not use commute back to home, also had a moderate to severe level of anxiety. The 172 (31%) participants who didn't have any physical activity in their lives, had moderate to severe level of anxiety. There were 101 (18%) participants who play video gaming, had a moderate to severe level of anxiety. The 148 (28%) participants who were suffering moderate to severe level of anxiety, had an at least one family member who does smoke.

## Bivariate analysis (screening predictor variables for model building)

As per our surveyed data, Weight, Ethnicity, and the number of households have an insignificant predictive variable with the level of Anxiety. On the other hand age (COR:0.057, p-value 0.171), Gender (COR: -0.323, p-value: 0.00, strict parents (COR: 0.035, p-value: 0.02), having tuition (COR: 0.109, p-value: 0.011), physical activities (COR: -0.059, p-value: 0.00), Smoker at home (COR: 0.006, p-value: 0.00), Lunch (COR:0.033, p-value: 0.01), School commute (COR:0.729, p-value: 0.003) all are significant predictive values which also represented in Table 2 from a supplementary file, which show the bi-variate analysis to select the considerable

**Table 1. Descriptive statistical analysis of participants.** Demographic Characteristics of the Study Sample (N = 544).

| variables | co-variables | Total | Moderate to severe Anxiety | Minimal to mild Anxiety |
|---|---|---|---|---|
| | | n (%) | n (%) | n (%) |
| **Gender** | Male | 231(42) | 33(7) | 99(18) |
| | Female | 313(57) | 140(26) | 272(50) |
| **Age** | 10 years -12 years | 60(11) | 16(3) | 44(8) |
| | 13 years—15 years | 350(64) | 109(20) | 241(44) |
| | 16 years– 19 years | 134(25) | 48(9) | 86(16) |
| **Ethnicity** | Urdu speaking | 392(72) | 105(20) | 287(53) |
| | Punjabi | 68(12) | 21(3) | 47(8) |
| | Sindhi | 12(2) | 9(2) | 3(0) |
| | Baloch | 9(2) | 7(1) | 2(0) |
| | Hazara | 4(1) | 2(0) | 2(0) |
| | Pathan | 59(11) | 29(6) | 30(6) |
| **Lunch** | Homemade food | 55(10) | 20(3) | 35(6) |
| | Canteen food | 489(90) | 153(28) | 336(62) |
| **No. of households** | I to 3 | 12(2) | 2(0) | 10(2) |
| | 4 to 6 | 295(54) | 88(16) | 207(39) |
| | 7 to 10 | 187(34) | 69(12) | 131(4) |
| | More than 10 | 50(9) | 14(2) | 16(3) |
| **Strict parent** | Father | 353(65) | 108(20) | 245(45) |
| | Mother | 191(35) | 65(12) | 126(23) |
| **School commutation** | Yes | 208(38) | 82(15) | 126(24) |
| | No | 336(62) | 91(16) | 245(25) |
| **Tuition** | Yes | 225(41) | 58(10) | 167(31) |
| | No | 319(59) | 115(21) | 204(38) |
| **Physical activities** | Yes | 194(35) | 57(10) | 137(25) |
| | No | 350(64) | 172(31) | 178(33) |
| **Video gaming** | Yes | 336(62) | 101(18) | 235(43) |
| | No | 208(38) | 72(13) | 136(25) |
| **Smoking at home** | Yes | 467(86) | 148(28) | 319(59) |
| | No | 77(14) | 25(5) | 52(10) |

**Table 2. Bivariate analysis between predictive variable with level of anxiety.**

| Variables | COR | p-value |
|---|---|---|
| **Age** | 0.057 | 0.171 |
| **Weight** | 0.091 | 0.25 |
| **Gender** | -0.323 | 0.00 |
| **Strict parents** | 0.035 | 0.02 |
| **Tuition** | -0.109 | 0.011 |
| **Physical activities** | -0.059 | 0.00 |
| **Smoking at home** | 0.006 | 0.00 |
| **Lunch** | 0.033 | 0.01 |
| **School commute** | 0.729 | 0.003 |

predictors with the level of Anxiety. Factors associated with anxiety level among school-going adolescents.

## Multivariate analysis between significant predictors and anxiety

Table 3, from a supplementary file, demonstrates the association between the levels of Anxiety with significant predictors. The odds of having moderate to severe levels of anxiety among the females were 1.22 times the odds of anxiety in males (p-value<0.05, 95% CI 0.675–3.702). In addition, the odds of having moderate to severe level of anxiety among adolescents who consume canteen food for lunch were 1.52 times the odds of anxiety among those who consume home food in lunch (p-value< 0.05, 95% CI 0.672–2.375). The odds of having moderate to severe level of anxiety among those adolescents who had at least one smoker family member were 1.398 times the odds of having anxiety who don't have anyone at home who smoke (p-value<0.05, 95% CI 0.732–2.668). Moreover, the odds of having moderate to severe level of anxiety among students who had strict fathers were 1.01 times the odds of having anxiety among students who had strict mother at home (p-value<0.05, 95% CI 0.645–1.582). Similarly, the odds of having moderate to severe level of anxiety among adolescents with sedentary lifestyles were 1.037 times the odds of having anxiety among adolescents with an active lifestyle (p-value<0.05, 95% CI 0.645–1.668).

## Anthropometric measurements of male vs female adolescents

Table 4 shows the anthropometric measurements of each gender of adolescents of the studied population in which 38% and 16% belonged to overweight and Obese class 1, respectively, having blood pressure in the 95th percentile ranging from (130+4/ 85+3mmHg) in male adolescents and 135+4/89+6mmHg) in females was measure among the studied population. Details of each gender age-wise are mentioned in Table 4.

## Interaction or confounding variables

All plausible interaction and confounding effects have been assessed. However, no interaction and confounding effect was found among study variables."

**Table 3. Multivariate analysis between significant predictors and anxiety.**

| Variable | Co-variable | Moderate to Severe Anxiety | Minimal to Mild Anxiety | OR | p-value |
|---|---|---|---|---|---|
|  |  | (n) | (n) | (95% CI) |  |
| Gender | Female | 140 | 272 | 1.220 (0.675–3.702) | 0.02 |
|  | Male | 33 | 99 | 1.00 |  |
| School commute | Yes | 87 | 121 | 1.040 (0.59–1.495) | 0.00 |
|  | No | 96 | 240 | 1.00 |  |
| Lunch | Canteen food | 161 | 328 | 1.520 (0.672–2.375) | 0.01 |
|  | Homemade food | 22 | 33 | 1.00 |  |
| Smoking at home | Yes | 154 | 313 | 1.398 (0.732–2.668) | 0.00 |
|  | No | 29 | 48 | 1.00 |  |
| Tuition | Yes | 65 | 160 | 1.070 (0.661–1.731) | 0.00 |
|  | No | 118 | 201 | 1.00 |  |
| Strict parents | Father | 114 | 239 | 1.010 (0.645–1.582) | 0.04 |
|  | Mother | 69 | 121 | 1.00 |  |
| Physical activities | Yes | 84 | 188 | 1.037 (0.645–1.668) | 0.03 |
|  | No | 99 | 173 | 1.00 |  |

**Table 4. Anthropometric measurements of male vs female adolescents of vulnerable population of Karachi, Pakistan.**

| Age (Years) | Anthropometric Data for Male | | | | Anthropometric Data for Female | | | |
|---|---|---|---|---|---|---|---|---|
| | Male (n) | BMI (kg/m2) | Bp (mmHg) | | Female (n) | BMI* (kg/m2) | Bp (mmHg)** | |
| | | | Systolic (SD) | Diastolic (SD) | | | Systolic (SD) | Diastolic (SD) |
| 10 | 5 | 21±0.9 | 100±7 | 72±3 | 8 | 21±1.9 | 110±4 | 77±2 |
| 11 | 5 | 17±1.3 | 85±5 | 60±2 | 19 | 15±3.0 | 90±2 | 55±3 |
| 12 | 3 | 18±0.5 | 84±8 | 55±5 | 20 | 28±2.9 | 125±4 | 88±1 |
| 13 | 35 | 20±2 | 120±3 | 70±2 | 43 | 17±2.7 | 90±4 | 50±8 |
| 14 | 47 | 19±1.9 | 118±6 | 79±5 | 56 | 23±2.7 | 126±5 | 85±7 |
| 15 | 77 | 24±1.0 | 130±4 | 85±3 | 92 | 23±1.5 | 135±4 | 89±6 |
| 16 | 20 | 16±1.8 | 98±2 | 58±3 | 30 | 16±1.9 | 80±7 | 49±4 |
| 17 | 21 | 26±3.5 | 140±7 | 100±5 | 24 | 26±1.7 | 150±7 | 90±1 |
| 18 | 18 | 22±0.5 | 115±5 | 75±2 | 21 | 28±2.9 | 135±3 | 105±8 |

BMI is Body mass index, BP is blood pressure

## Discussion

Nowadays, mental health is one of the most challenging public health concerns globally [17], which is increasing day by day among children and adolescents, especially in adolescents [18] who are reaching puberty age [19].

In our study, we found that Females show they are more Anxiety than male adolescents, approximately 57% in our studied population this may be due to hormonal changes, hormonal imbalance [20], metabolic syndrome [21] which is more prone in female than male [21], this has also been proven in the previous study conducted in Italy, in which the female gender showed 54.6% anxiety which is around our findings [22]. This has been proven so many other studies as well, such as the era between 2010–2014; the UK concluded that approximately 68% of young girls of aged 13-15years showed a rise in suicidal tendency from 45.9 per 10 000 in 2011 to 77.0 per 10 000 due to Anxiety and depression [23].

In our analysis, it has been shown that parenting style influence on increase or decrease of anxiety level among children and adolescents. Strict parenting style with more restrictions, rude behaviors, no listening attitude also plays a significant role in developing this may be because of the socio-economic status of slum dwellers where the parents spend most of their time in earning the bread and butter for the family so they have little or no time for their children. Anxiety among youngsters, as discussed by some researchers from Spain in 2019 [24]. In groups, researchers from Brazil and China in 2020 claimed in their studies that Parents' stress levels, illiteracy, or low educational status are the main reasons for developing mental health issues among their children and adolescents [25, 26]. Our healthcare system needs to educate and counsel parents when expecting their children. This and pregnant mothers' health and clinical screening should be mandatory. There are several reasons which influence Anxiety on gender; for instance, closed family systems in tribal increase the toll of mental health issues in their females or due to nervousness and low confidence in the population as studied by a group of researchers Liu R. et al. from China in 2021 during COVID pandemic [27], in addition to this regional culture and attitude toward no freedom given to their females causing the development of stress and other mental health complications also are the main reason of Anxiety in females as discussed by Osama M. et al. from Egypt in 2019 [28].

In our study, we found that limited or no Physical activity and sedentary lifestyles also support and develop Anxiety among adolescents, which has also been discussed in one of the

studies conducted in the UK by Kandola et al., which stated that adolescents who spend maximum time inactive show more Anxiety in their later life [29]. In addition, as per our study, we found that adolescents who spend time gaming on computers, laptops, or mobile also support an increased magnitude of stress and Anxiety among adolescents. More screen timing ultimately led them to sleep less which is not good for their mental well-being. They will be less active and give no or less attention to their studies. A previous study in China supports this finding conducted in the year 2019 in which they found a direct association between mobile gaming and Anxiety [30]. On the other hand, our study also supports that those who avail the transport instead of walking increase moderate to severe Anxiety; this may be because of developing Anxiety about reaching schools on time due to the extended distance from home to school. This has also been shown and proven in one of the California-based studies in 2013 in which they discussed that school distance directly impacts the Anxiety of children and adolescents [31]. In addition, Anxiety due to fear of surroundings also affects school-going adolescents who interact on their way to school, as discussed in another study conducted in Australia in 2016 [32].

Our study also shows that those adolescents who belong to smoking families show moderate to severe Anxiety which might be due to the adolescents can adopt the habit of smoking from their family for having fun which is not beneficial for their health. This has also shown in one of the studies conducted in Turkey in the year 2021, in which Ayran G et al. (2022) found that those who had other smokers in family shown severe kind of Anxiety and also nicotine dependency in early ages of their life [33]. Similarly, a group of Researchers from Qatar, Ben Brik A. et al. (2022), shared their recent study outcomes that non-smoker parents with active and healthy lifestyles show less Anxiety among their children [34].

Food intake has an imperative impact on Anxiety among school-going adolescents in our collected data. Having homemade food in lunch maintain the level of nutrients in the body. Those adolescents who rare having lunch from home have minimum level of anxiety than those who are having unhealthy food from the canteen. This has also been shown in a study by Khosravi M. et al. (2015) from Iran, which found that unhealthy dietary patterns significantly impact diet depression among children and adolescents [35]. Similarly, USA investigators Crawford GB et al. (2011) also concluded the same observation that junk food consumption causes higher depression among youngsters [36].

In our analysis, those participants who are taking tuition show a significant level of moderate to severe Anxiety compared to those who are availing of this opportunity. Social phobia also plays a significant part in adolescents' anxiety. Adolescents hesitate and frighten to interact people. Due to which they don't want to take shallow educations which effect their studies. Shallow education helps children to be able to learn more diligently than those who are learning by themselves without taking the help of others. One of the studies from India in 2018 also supports that secondary school-going children and adolescents show Anxiety and academic stress of 45%, who were not going to tuition centers along with the schools [37].

In our analysis, 38% to 16% belong to overweight and class 1 obesity as per their height-to-weight ratio. Weight play an imperative role of controlling so many functions in human body. Overweight or near to obese or obese people are more prone of the anxiety [38] and depression this may because of many reasons but as of yet, there does not seem to be a clear or direct link between obesity and either anxiety or depression [39]. Some studies show a link between adolescent obesity and depression in early adulthood [40], whereas others show a link between adolescent severe depression and a higher BMI in adulthood, especially in females [41]. Weight reduction may significantly alleviate depression, and there is a clear correlation between obesity and depression in those who are very obese [42]. Similarly, the same pattern was also shown in a study conducted by Dietrich et al. [43], which found that BP. Heart rate and BMI

play a significant role in anxiety levels. However, on the other hand, Narmandakh A et al. (2021) found no association of such parameters with anxiety levels [44].

## Strength of the study

This study will measure the anxiety disorders in understudied school-going children and adolescents of low-middle stratification in Karachi, Pakistan. This study will add the edge to mental health well-being, where very few studies have been published. By doing this kind of study, we can solve different mental issues related to adolescents in a highly vulnerable state because Patients with mild cognitive Anxiety might be particularly vulnerable and lead to anxiety disorder.

## Limitation of the study

The nature of the study itself is the major limitation. The data were collected within one month time period and from 5 schools. It is not representing the whole population. It could have been more significant if we considered two or three months of data and approached more schools due to management reluctance, inadequate exposure to such studies, and school access to too much work. Many adolescents hesitate to share their issues due to misjudgments. Also, some parents should have prevented us from collecting data from their children.

## Future directions

We recommended expanding research on this sensitive topic so that the probability of successful versions of anxiety interventions in clinical practice can be possible. Moreover, through this study, further studies can be done concerning other clinical symptoms, such as the association of Anxiety with the sleeping pattern, eating disorders, and performance of children and adolescents in their schools, leading to depression or further suicidal ideation later in life.

Despite the fact that our findings are from a cross-sectional study, the significant relationships between mental health issues, unhealthy behaviours, and obesity that we discovered point to the need for programmes to simultaneously improve the physical and mental well-being of individuals to be developed and implemented.

## Conclusion

Our study found that adolescents suffer moderate to severe anxiety levels at their growing age, especially those who reach puberty. If negligence, this may worsen to more severe, leading to depression and panic attacks with time. More studies should also be conducted to assess adolescents' Anxiety and other cognitive mental health problems. In context with the shallow consideration towards its assessment and treatment, the high occurrence of anxiety problems recommends an essential method to identify and publicize analytically validated treatments in the primary care setting in response to these problems.

## Acknowledgments

The authors would like to thank the Management of SINA Health and Welfare Education Trust, especially CEO Ms. Ambareen Kazim Thompson, and all school's management who consented to this study and gave full cooperation and helped us in data collection.

## Author Contributions

**Conceptualization:** Tooba Seemi.

**Data curation:** Tooba Seemi, Hira Naeem, Farhat ul Ain Naeem.

**Formal analysis:** Sana Sharif.

**Investigation:** Hina Sharif, Sana Sharif.

**Methodology:** Hina Sharif, Farhat ul Ain Naeem, Zoya Fatima.

**Resources:** Hina Sharif, Hira Naeem.

**Software:** Sana Sharif.

**Supervision:** Hina Sharif.

**Validation:** Sana Sharif.

**Writing – original draft:** Tooba Seemi, Hira Naeem.

**Writing – review & editing:** Hina Sharif, Zoya Fatima.

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
