## [Decision Letter · Decision Letter 0]

22 Mar 2023

PONE-D-23-04889Anxiety levels among school-going adolescents in peri-urban areas of Karachi, PakistanPLOS ONE

Dear Dr. Sharif,

Thank you for submitting your manuscript to PLOS ONE. After careful consideration, we feel that it has merit but does not fully meet PLOS ONE’s publication criteria as it currently stands. Therefore, we invite you to submit a revised version of the manuscript that addresses the points raised during the review process.

We look forward to receiving your revised manuscript.

Kind regards,

Sheikh Mohd Saleem, MBBS, MD

Academic Editor

PLOS ONE

Journal Requirements:

Additional Editor Comments:

Dear Authors,

Thank you for submitting your manuscript titled "Anxiety levels among school-going adolescents in peri-urban areas of Karachi, Pakistan" to PLOS. I have carefully reviewed the manuscript and would like to provide feedback on the areas that require improvement.

Firstly, the abstract requires a clear rationale for the study and a brief summary of the key findings and strengths of the study. The introduction section should be rewritten to better emphasize the background of the study and the need for it.

In the objectives section, the authors should rephrase and condense the eight objectives into a single comprehensive objective that encapsulates the main aim of the study. The methods section needs to be revisited and made more specific. The term "self-generated" should be clarified, and the authors should provide information on how the questionnaire was developed and validated, including details about instrument calibration.

The authors should also revise the findings section to present the results in a more coherent and comprehensive manner. The statistical part needs attention. The conclusion and recommendations section needs to be rewritten to clearly summarize the main findings of the study and provide practical recommendations for policymakers and researchers.

Lastly, the authors need to edit the entire manuscript for language and grammar. The language used should be clear and concise, and the grammar should be accurate.

I suggest that the authors carefully review and revise the manuscript in light of these comments and suggestions. Once the revisions have been made, please resubmit the manuscript.

Thank you for your submission, and I look forward to receiving your revised manuscript.

Best regards,

Reviewers' comments:

Reviewer's Responses to Questions

**Comments to the Author**

1. Is the manuscript technically sound, and do the data support the conclusions?

Reviewer #1: Partly

Reviewer #2: Partly

2. Has the statistical analysis been performed appropriately and rigorously? 

Reviewer #1: Yes

Reviewer #2: Yes

3. Have the authors made all data underlying the findings in their manuscript fully available?

Reviewer #1: No

Reviewer #2: Yes

4. Is the manuscript presented in an intelligible fashion and written in standard English?

Reviewer #1: No

Reviewer #2: No

5. Review Comments to the Author

Reviewer #1: The study planned is of great importance, but not presented in an intelligent fashion.

1. Too many objectives mentioned , which can be addressed in two lines.

2. Unnecessary prolonged calculation of sample size. The calculated sample size with 50% expected rate (conservatively) produces a sample size of 385, with given confidence level and margin of error, and this sample size achieved even produces margin of error precisely is 0.042.

3. Analysis is done with logic but presented poorly and incorrectly. The severe anxiety should be the first column and mild/none the second, and each factor tested for association should also be presented in a manner that the risk factor should come in first row and other in second.

4. The significance levels seems to be incorrect. For example as an Adjusted Odds Ratio of 1.07 with interval of 0.661 - 1.731 is always insignificant, and it can never have a p-value <0.05.

The authors need to consult some professional statistician who can guide them on these aspects and need to revise the manuscript with better professional approach.

Reviewer #2: Anxiety levels among school-going adolescents in peri-urban areas of Karachi, Pakistan

Abstract

1. Please include rationale of the study and what this study adds to the exiting evidence.

2. Methods: “self-generated”???

3. Please revisit the methods section and make it more specific

4. Please rewrite the abstract

What this study adds: please rewrite the entire section to highlight the key findings and strengths of the study. Please revisit this text mentioned in the manuscript. “Pakistan is facing a serious challenge in ensuring the enrolment and retention of children and adolescents at schools [1]. There is a dire need to ensure such an education system in the country.”

Introduction: please rewrite and rationale for the study needs to be inbuilt within the background section

Objectives: The authors have mentioned 8 objectives in the manuscript. Please mention a comprehensive objective and rewrite.

Methods: the authors mention “systemic random sample” in sample size section (which is refereed as data size in the manuscript). Please revise to systematic.

Please clarify self-made form.

How was the questionnaire developed and validated?

Please mention about calibration of instruments as well.

Findings need to be rewritten and presented in a comprehensive manner.

Please rewrite conclusion and recommendations section.

The authors need to edit the entire manuscript for language and grammar.

6. PLOS authors have the option to publish the peer review history of their article (what does this mean?). If published, this will include your full peer review and any attached files.

Reviewer #1: No

Reviewer #2: No

---

## [Author Response · Author response to Decision Letter 0]

23 Jun 2023

Thanks a lot, dear Editor and respected reviewers. It will always be informative to get such reviews which we can improve in our current and next research articles. All the changes have been revised; you will see them in the track change file.

 RESPONSE TO REVIEWER & EDITOR 

1 1. Please include rationale of the study and what this study adds to the exiting evidence

 Thanks reviewer, we have added in revised manuscript 

 2. Methods: “self-generated”??? we have replaced the present from the following statement

Self-designed structured pre-tested questionnaires.

 3. Please revisit the methods section and make it more specific

 We have revised the method section and it will shown in track changes file

 4. Please rewrite the abstract The abstract we have changed from previous to submitted one and made it more clear and specific to the study point of view

 5. What this study adds: please rewrite the entire section to highlight the key findings and strengths of the study. Please revisit this text mentioned in the manuscript. “Pakistan is facing a serious challenge in ensuring the enrolment and retention of children and adolescents at schools [1]. There is a dire need to ensure such an education system in the country.” We have revised the rationale of the study 

 6. Introduction: please rewrite and rationale for the study needs to be inbuilt within the background section We have revised the introduction part and cite with supporting literature.

 7. Objectives: The authors have mentioned 8 objectives in the manuscript. Please mention a comprehensive objective and rewrite. We have concise from 8 to 4 objectives

 8. Methods: the authors mention “systemic random sample” in sample size section (which is refereed as data size in the manuscript). Please revise to systematic. Thanks for pointed out this, we have changed in revised manuscript

 9. Please clarify self-made form we have replaced the present from the following statement

Self-designed structured pre-tested questionnaires.

 10. How was the questionnaire developed and validated? The questionnaire was developed with the help of psychologist and the statement also added regarding this in our revised manuscript

 11. Please mention about calibration of instruments as well. The questionnaire was developed with the help of psychologist and tested into 15 adolescents for validity. the statement also added regarding this in our revised manuscript

 12. Findings need to be rewritten and presented in a comprehensive manner. We have tried to revised and make it more comprehensive and related to the objectives

 13. Please rewrite conclusion and recommendations section. We have tried to revised and make it more comprehensive and related to the objectives

 14. The authors need to edit the entire manuscript for language and grammar. We have referred English grammar rectification tool to omit our mistakes

---

## [Decision Letter · Decision Letter 1]

17 Jul 2023

PONE-D-23-04889R1Anxiety levels among school-going adolescents in peri-urban areas of Karachi, PakistanPLOS ONE

Dear Dr. Sharif,

Thank you for submitting your manuscript to PLOS ONE. After careful consideration, we feel that it has merit but does not fully meet PLOS ONE’s publication criteria as it currently stands. Therefore, we invite you to submit a revised version of the manuscript that addresses the points raised during the review process.

We look forward to receiving your revised manuscript.

Kind regards,

Sheikh Mohd Saleem, MBBS, MD

Academic Editor

PLOS ONE

Journal Requirements:

Reviewers' comments:

Reviewer's Responses to Questions

**Comments to the Author**

1. If the authors have adequately addressed your comments raised in a previous round of review and you feel that this manuscript is now acceptable for publication, you may indicate that here to bypass the “Comments to the Author” section, enter your conflict of interest statement in the “Confidential to Editor” section, and submit your "Accept" recommendation.

Reviewer #2: (No Response)

Reviewer #3: All comments have been addressed

2. Is the manuscript technically sound, and do the data support the conclusions?

Reviewer #2: Yes

Reviewer #3: Yes

3. Has the statistical analysis been performed appropriately and rigorously? 

Reviewer #2: Yes

Reviewer #3: Yes

4. Have the authors made all data underlying the findings in their manuscript fully available?

Reviewer #2: Yes

Reviewer #3: Yes

5. Is the manuscript presented in an intelligible fashion and written in standard English?

Reviewer #2: Yes

Reviewer #3: Yes

6. Review Comments to the Author

Reviewer #2: Review:

Anxiety levels among school-going adolescents in peri-urban areas of Karachi, Pakistan

Overall comments:

1.Please re-write the conclusion in the abstract section.

2.Please edit the manuscript for language and grammar.

3.Please rewrite the ‘what this study adds’ section in the revised manuscript.

Introduction:

1.Please consider embedding rationale and objectives into the introduction section.

2.The objectives could be rewritten comprehensively.

Methods:

1.‘A systemic random sample’- please revise it to systematic and mention how systematic sampling was undertaken in the text.

2.‘we assumed the majority as 50% through a literature search,’: please check and cite references for the literature review undertaken

3.Under sample size calculation: The text mentions ‘Patient sample size’- who are the patients here?

4.Pilot study was undertaken among 15 students on what basis? At least 10-20% of the sample should have been considered.

Discussion:

1.Authors may consider adding implications of the study

Strengths:

1.‘By doing this kind of study, we can solve different mental issues in adolescents of a highly vulnerable state because Patients with mild cognitive Anxiety might be particularly vulnerable and lead to anxiety disorder’- how ??

Reviewer #3: The manuscript is a eloquent read. The authors have addressed all the previous comments which were asked by previous reviewers. i found grammatical and language mistakes and which needs to be corrected before publication.

7. PLOS authors have the option to publish the peer review history of their article (what does this mean?). If published, this will include your full peer review and any attached files.

Reviewer #2: No

Reviewer #3: No

---

## [Author Response · Author response to Decision Letter 1]

19 Jul 2023

Reviewer 2 Query

Reviewer Query Response from the authors

1.Please re-write the conclusion in the abstract section.

 Thank you for highlighting this, we have revised the conclusion please see pg 2

2.Please edit the manuscript for language and grammar.

 Noted!, We have considered the English editing service to omit the grammatical and language errors

3.Please rewrite the ‘what this study adds’ section in the revised manuscript.

Introduction:

 Thank you for highlighting this, we have revised the conclusion please see pg 3

1.Please consider embedding rationale and objectives into the introduction section.

 Thank you for highlighting this, we have revised the conclusion please see pg 3-4

2.The objectives could be rewritten comprehensively.

 Thank you for highlighting this, we have revised the conclusion please see pg 4

Methods:

1.‘A systemic random sample’- please revise it to systematic and mention how systematic sampling was undertaken in the text.

 Thank you for highlighting this, we have revised the conclusion please see pg 4

2.‘we assumed the majority as 50% through a literature search,’: please check and cite references for the literature review undertaken

 Thank you for highlighting this, we have revised the conclusion please see pg 4

3.Under sample size calculation: The text mentions ‘Patient sample size’- who are the patients here?

 Apology for the typo error we have rectified and replaced by students

4.Pilot study was undertaken among 15 students on what basis? At least 10-20% of the sample should have been considered.

 Apology for the typo error! We have added 15% 

Discussion:

1.Authors may consider adding implications of the study

 Thank you we have tried to add some implications for this study

Strengths:

1.‘By doing this kind of study, we can solve different mental issues in adolescents of a highly vulnerable state because Patients with mild cognitive Anxiety might be particularly vulnerable and lead to anxiety disorder’- how ?? Due to screening the anxiety level in students we can able to know if pupil have anxiety or not. After screening we can do further measures to prevent this state after screening and highlighted it to their parents so that further treatment can be done.

Reviewer 3

Reviewer Query Response from the authors

The manuscript is a eloquent read. The authors have addressed all the previous comments which were asked by previous reviewers. i found grammatical and language mistakes and which needs to be corrected before publication. Thank you for the feedback! 

We have considered the English editing service to omit the grammatical and language errors

---

## [Editor Report · Decision Letter 2]

20 Jul 2023

PONE-D-23-04889R2Anxiety levels among school-going adolescents in peri-urban areas of Karachi, PakistanPLOS ONE

Dear Dr. Hina,

Thank you for submitting your manuscript to PLOS ONE. After careful consideration, we feel that it has merit but does not fully meet PLOS ONE’s publication criteria as it currently stands. Therefore, we invite you to submit a revised version of the manuscript that addresses the points raised during the review process.

We look forward to receiving your revised manuscript.

Kind regards,

Sheikh Mohd Saleem, MBBS, MD

Academic Editor

PLOS ONE

Journal Requirements:

Additional Editor Comments (if provided):

ABSTRACT

Overall, the introduction and findings of the abstract provide a good overview of the study on anxiety among school-going adolescents in peri-urban areas of Karachi, Pakistan. However, there are some areas where improvements could be made:

1.Clarity in Objectives: The objective is stated but could be made clearer by specifically mentioning what the study aims to achieve, such as understanding the prevalence of anxiety, identifying risk factors, or assessing the impact of anxiety on adolescents' well-being.

2.Methodology: Provide more details about the sampling method and how the schools were selected. It would be helpful to mention the total number of schools approached and the criteria used for inclusion in the study.

3.Missing Information: The abstract lacks information on the response rate or participation rate of the students in the study. Including this data would give readers an idea of the representativeness of the sample.

4.Results: The findings mentioned in the abstract should be more specific. For example, provide the total number of participants who had moderate to severe anxiety symptoms, and clearly state the prevalence rates among males and females.

5.Statistical Significance: If possible, mention whether the associations between variables and anxiety levels were statistically significant to add more credibility to the findings.

6.Implications: The abstract could benefit from briefly discussing the potential implications of the study's findings. How could the results help in developing intervention programs or improving mental health services for adolescents in peri-urban areas?

7.Conclusion: The conclusion is clear in summarizing the key findings, but it could be strengthened by providing a brief recommendation for future research or practical implications for addressing adolescent anxiety.

Introduction

8.The introduction provides a clear overview of the importance of studying anxiety among adolescents and the lack of research in peri-urban areas of Karachi, Pakistan.

9.The introduction successfully highlights the global prevalence of anxiety disorders among adolescents, providing relevant statistics.

10.However, it would be helpful to include the specific sources for the cited statistics to enhance the credibility of the information.

11.There seems to be repetition of information regarding the GAD-7 scale and its psychometric qualities. Consider consolidating this information into a concise paragraph.

12.The mention of the recent studies conducted in the USA and Europe using the GAD-7 scale strengthens the case for using this tool for anxiety screening . However, it might be beneficial to provide a brief summary of their key findings.

13.The rationale for the study is well-explained, focusing on the lack of mental health screening for school-going adolescents in peri-urban areas of Karachi, Pakistan, and the specific challenges they face.

14.Consider including more specific information about the challenges faced by adolescents in peri-urban areas of Karachi, such as economic difficulties, lack of mental health resources, and the impact of the COVID-19 outbreak.

15.The objectives of the study are clearly stated, addressing the different factors that may contribute to anxiety levels among school-going adolescents .

16.It would be beneficial to include a brief overview of the methodology, such as the sample size, age group, data collection methods, and statistical analysis techniques to be used.

17.Ensure that the use of references is consistent throughout the introduction.

18.Overall, the introduction provides a solid foundation for the study, highlighting the significance of addressing anxiety among school-going adolescents in peri-urban areas. By addressing the points mentioned above, the introduction can be further improved for a more comprehensive and compelling overview of the research.

Methodology

19.The study site and sample selection process are well-explained, providing relevant information about the three selected slum areas and the rationale behind their selection.

20.The study's ethics approval and consent process are clearly described, ensuring that participants' identities remain confidential.

21.The justification for the sample size calculation is provided, considering a prevalence rate of 50% and a 20% no response rate. However, it would be helpful to include a brief explanation of the assumptions made in choosing these values.

22.The inclusion and exclusion criteria are clearly stated, defining the age range, socio-economic status, and consent requirements for participation.

23.The description of the data collection tools, including the demographic form and the GAD-7 scale, is well-presented, providing clarity on what information is being collected.

24.The validation process of the self-designed structured questionnaire through a pilot survey is a good approach to ensure the validity of the instrument.

25.The data collection process is well-outlined, mentioning the schools' selection, translation of the questionnaire, and the involvement of researchers in administering the survey.

26.The anthropometric measurements section provides details on how blood pressure measurements were taken and validated, which adds to the study's credibility.

27.The statistical analysis plan is comprehensive, covering data review, descriptive statistics, bivariate analysis, and multivariate logistics regression to identify significant predictors of anxiety levels.

28.It would be helpful to include a brief mention of the data analysis software used (SPSS version 24) and any statistical tests that will be employed.

29.Consider adding information about any measures taken to ensure data quality and minimize biases during data collection.

30.Ensure that all figures and tables mentioned in the text are included and appropriately labeled.

31.Overall, the methodology section is well-structured and provides sufficient details for understanding the study's design and data collection process. By addressing the points mentioned above, the methodology section can be further enhanced for a comprehensive and robust study.

Results

32.The introduction of the results section provides a clear summary of the participant recruitment process, with 700 participants approached and 544 participants included for final analysis.

33.Table 1 is mentioned, which shows socio-demographic analysis. However, it would be more informative if the key findings from Table 1 are briefly described in the text, highlighting the relevant demographic characteristics of the participants.

34.The use of odds ratios (OR) and confidence intervals (CI) to demonstrate the association between anxiety levels and various factors is appropriate. However, it is essential to interpret these results carefully, considering the wide confidence intervals and the potential for non-significant associations.

35.The results describe the association between anxiety levels and factors such as gender, food intake, presence of a smoker in the family, and parenting style. However, it would be more informative to present the actual values of OR and CI, not just stating whether they are significant or not.

36.Provide more clarity on the interpretation of the OR values. For example, an OR of 1.010 suggests a very minimal difference, so it is important to specify the practical significance of such results.

37.When mentioning percentages, ensure that the total sample size for each gender is specified for better context and interpretation.

38.It would be helpful to include additional statistical measures, such as p-values, to determine the significance of the observed associations.

39.The description of Table 4, showing anthropometric measurements, is brief and lacking in interpretation. Consider discussing any noteworthy findings or patterns in these measurements among adolescents.

40.Overall, the results section is informative, but additional details and interpretation of the findings would enhance its clarity and significance.

41.Consider organizing the results section with subheadings to make it more structured and reader-friendly, especially when referring to different tables.

Discussion

42.The discussion starts with a relevant and concise introduction that highlights the significance of mental health concerns among children and adolescents globally.

43.The first point discusses the higher prevalence of anxiety in females, which is supported by previous studies. However, it would be beneficial to elaborate on potential reasons for this gender difference in anxiety, such as hormonal factors, societal expectations, or coping mechanisms.

44.The discussion on parenting style and its influence on anxiety is well-presented, citing studies from different countries. To strengthen this point, consider discussing specific parenting practices associated with higher anxiety levels and potential interventions to promote healthier parenting styles.

45.When mentioning previous studies, ensure accurate referencing by providing complete citation details, including authors, year, and journal or publication.

46.The discussion on sedentary lifestyles, gaming, and transportation as factors influencing anxiety is informative. Consider discussing the underlying mechanisms behind these associations, such as increased screen time affecting sleep patterns or reduced physical activity affecting stress management.

47.To strengthen the discussion on adolescents from smoking families, provide further insights into the potential impact of secondhand smoke exposure on anxiety levels in adolescents. Discuss possible interventions to support families in reducing smoking and promoting a healthier environment for adolescents.

48.The impact of food intake on anxiety is well-explained, supported by relevant studies. To enhance this point, discuss the role of specific nutrients and dietary patterns in influencing mood and anxiety in adolescents.

49.The discussion on tuition and academic stress is valuable. Expand on potential reasons why tuition attendance may be associated with higher anxiety levels. Discuss strategies to promote academic support while minimizing stress among students.

50.The comparison of BMI and anxiety levels is relevant. However, provide a more nuanced analysis of the relationship, considering potential confounding factors, such as age, gender, and lifestyle habits. Discuss how BMI changes over time may influence anxiety outcomes.

51.Consider adding a final section summarizing the key findings and implications of the study. Discuss how the results contribute to the existing body of knowledge on adolescent anxiety and suggest potential interventions or areas for further research.

52.Overall, the discussion is well-structured, and the inclusion of supporting studies adds credibility to the presented findings. To enhance the section, provide more in-depth analysis and interpretation of the results, while also highlighting potential avenues for future research and practical implications for mental health interventions.
---

## [Author Response · Author response to Decision Letter 2]

26 Jul 2023

Clarity in object: The objective is stated but could be made clearer by specifically mentioning what the study aims to achieve, such as understanding the prevalence of anxiety, identifying risk factors, or assessing the impact of anxiety on adolescents' well-being. Thank you for your guidance. We rectified this in our manuscript on page no. 02

Methodology: Provide more details about the sampling method and how the schools were selected. It would be helpful to mention the total number of schools approached and the criteria used for inclusion in the study. We have revised the methodology and add the sampling procedure to clear the details. Thank you! For pointing out this.

Missing Information: The abstract lacks information on the response rate or participation rate of the students in the study. Including this data would give readers an idea of the representativeness of the sample. We have added the missing information in abstract now. Thank you!

Results: The findings mentioned in the abstract should be more specific. For example, provide the total number of participants who had moderate to severe anxiety symptoms, and clearly state the prevalence rates among males and females. Thank you for pointing out this. We modified this in our manuscript on page no: 02

Statistical Significance: If possible, mention whether the associations between variables and anxiety levels were statistically significant to add more credibility to the findings. We have added the lines for associations in the abstract. Thank you!

Implications: The abstract could benefit from briefly discussing the potential implications of the study's findings. How could the results help in developing intervention programs or improving mental health services for adolescents in peri-urban areas? Thank you for pointing out this. We modified this in our manuscript on page no: 02

Conclusion: The conclusion is clear in summarizing the key findings, but it could be strengthened by providing a brief recommendation for future research or practical implications for addressing adolescent anxiety. Thank you for your guidance. We modified this in our manuscript on page no: 02

The introduction provides a clear overview of the importance of studying anxiety among adolescents and the lack of research in peri-urban areas of Karachi, Pakistan. Thank you for your kind words and appreciation.

The introduction successfully highlights the global prevalence of anxiety disorders among adolescents, providing relevant statistics. We are thankful to you for this appreciation

However, it would be helpful to include the specific sources for the cited statistics to enhance the credibility of the information. Mis sana

There seems to be repetition of information regarding the GAD-7 scale and its psychometric qualities. Consider consolidating this information into a concise paragraph. Thank you for your guidance. We rectified this in our manuscript on page no. 03

The mention of the recent studies conducted in the USA and Europe using the GAD-7 scale strengthens the case for using this tool for anxiety screening. However, it might be beneficial to provide a brief summary of their key findings. Thank you for highlighting this. We rectified this in our manuscript on page no. 03

The rationale for the study is well-explained, focusing on the lack of mental health screening for school-going adolescents in peri-urban areas of Karachi, Pakistan, and the specific challenges they face. Thank you for your feedback.

Consider including more specific information about the challenges faced by adolescents in peri-urban areas of Karachi, such as economic difficulties, lack of mental health resources, and the impact of the COVID-19 outbreak. Thank you for highlighting this. We rectified this in our manuscript on page no. 03

The objectives of the study are clearly stated, addressing the different factors that may contribute to anxiety levels among school-going adolescents. Thanks for your feedback.

It would be beneficial to include a brief overview of the methodology, such as the sample size, age group, data collection methods, and statistical analysis techniques to be used. Thank you for pointing out this! We have revised the methodology section please see page# 5

Ensure that the use of references is consistent throughout the introduction. Thank you for highlighting this. We rectified that. Please see page no: 10. All the references numbers are in sequence now

Overall, the introduction provides a solid foundation for the study, highlighting the significance of addressing anxiety among school-going adolescents in peri-urban areas. By addressing the points mentioned above, the introduction can be further improved for a more comprehensive and compelling overview of the research Thank you for your feedback. We addressed the above points which you mentioned above.

Methodology. The study site and sample selection process are well-explained, providing relevant information about the three selected slum areas and the rationale behind their selection Thank you for your feedback

The study's ethics approval and consent process are clearly described, ensuring that participants' identities remain confidential. Thank you for your feedback

The justification for the sample size calculation is provided, considering a prevalence rate of 50% and a 20% no response rate. However, it would be helpful to include a brief explanation of the assumptions made in choosing these values. There is no such study in Pakistan especially in resource-poor settings. We have assumed the over all anxiety in 50% prevalence so that It would be easy for us to summarize the results with no bias

The inclusion and exclusion criteria are clearly stated, defining the age range, socio-economic status, and consent requirements for participation. Thanks for your kind feedback

The description of the data collection tools, including the demographic form and the GAD-7 scale, is well-presented, providing clarity on what information is being collected. Thank you for your feedback

The validation process of the self-designed structured questionnaire through a pilot survey is a good approach to ensure the validity of the instrument. Thank you for your appreciation 

The data collection process is well-outlined, mentioning the schools' selection, translation of the questionnaire, and the involvement of researchers in administering the survey. Thank you for your feedback

The anthropometric measurements section provides details on how blood pressure measurements were taken and validated, which adds to the study's credibility. Thank you for your feedback

The statistical analysis plan is comprehensive, covering data review, descriptive statistics, bivariate analysis, and multivariate logistics regression to identify significant predictors of anxiety levels. Thank you for your feedback

It would be helpful to include a brief mention of the data analysis software used (SPSS version 24) and any statistical tests that will be employed. Thank you! It has been mentioned in the first two lines under the heading of statistical analysis. Please refer page # 6

Consider adding information about any measures taken to ensure data quality and minimize biases during data collection. Thank you for pointing out this, we have add after result section in page # 7

Ensure that all figures and tables mentioned in the text are included and appropriately labeled. Thank you for highlighting this. We corrected this. Please see page no: 14,15,16,17

Overall, the methodology section is well-structured and provides sufficient details for understanding the study's design and data collection process. By addressing the points mentioned above, the methodology section can be further enhanced for a comprehensive and robust study. Thank you for your feedback. We rectified all the above-mentioned points.

Results: The introduction of the results section provides a clear summary of the participant recruitment process, with 700 participants approached and 544 participants included for final analysis. Thank you for your feedback.

Table 1 is mentioned, which shows socio-demographic analysis. However, it would be more informative if the key findings from Table 1 are briefly described in the text, highlighting the relevant demographic characteristics of the participants. Thank you for highlighting this point. We rectified this. Please check page no: 6

The use of odds ratios (OR) and confidence intervals (CI) to demonstrate the association between anxiety levels and various factors is appropriate. However, it is essential to interpret these results carefully, considering the wide confidence intervals and the potential for non-significant associations. We have modified the OR in the result section. Please see page # 6Thank you!

The results describe the association between anxiety levels and factors such as gender, food intake, presence of a smoker in the family, and parenting style. However, it would be more informative to present the actual values of OR and CI, not just stating whether they are significant or not. Thank you for pointing out this. We rectified this in our manuscript. Please see page # 6

Provide more clarity on the interpretation of the OR values. For example, an OR of 1.010 suggests a very minimal difference, so it is important to specify the practical significance of such results. We have modified the result section and made it more sound. Please refer page 6. Thank you!

When mentioning percentages, ensure that the total sample size for each gender is specified for better context and interpretation. We have added the total sample size of gender. Thank you!

It would be helpful to include additional statistical measures, such as p-values, to determine the significance of the observed associations. We have added the associations with p-value. Thank you!

The description of Table 4, showing anthropometric measurements, is brief and lacking in interpretation. Consider discussing any noteworthy findings or patterns in these measurements among adolescents Thank you for the suggestion! We have discussed in detail in page 9 last paragraph of discussion section.

Overall, the results section is informative, but additional details and interpretation of the findings would enhance its clarity and significance. T Thank you for your valuable comments, we have now modified the results section as per your instructions.

Consider organizing the results section with subheadings to make it more structured and reader-friendly, especially when referring to different tables. Thank you for highlighting this. We have added the headings as per your valuable instructions

Discussion: The discussion starts with a relevant and concise introduction that highlights the significance of mental health concerns among children and adolescents globally. Thank you for your feedback.

The first point discusses the higher prevalence of anxiety in females, which is supported by previous studies. However, it would be beneficial to elaborate on potential reasons for this gender difference in anxiety, such as hormonal factors, societal expectations, or coping mechanisms. Thank you for highlighting this. We rectified this in our manuscript. Please see discussion section at page no: 7

The discussion on parenting style and its influence on anxiety is well-presented, citing studies from different countries. To strengthen this point, consider discussing specific parenting practices associated with higher anxiety levels and potential interventions to promote healthier parenting styles. Thank you for your response. We have modified this. Please check page no: 7

When mentioning previous studies, ensure accurate referencing by providing complete citation details, including authors, year, and journal or publication. Thank you for pointing out this. We have rectified this in our manuscript. 

The discussion on sedentary lifestyles, gaming, and transportation as factors influencing anxiety is informative. Consider discussing the underlying mechanisms behind these associations, such as increased screen time affecting sleep patterns or reduced physical activity affecting stress management. Thank you for pointing out this. We have rectified this in our manuscript.

To strengthen the discussion on adolescents from smoking families, provide further insights into the potential impact of secondhand smoke exposure on anxiety levels in adolescents. Discuss possible interventions to support families in reducing smoking and promoting a healthier environment for adolescents. Thank you for pointing out this. We have rectified this in our manuscript. See page no: 8

The impact of food intake on anxiety is well-explained, supported by relevant studies. To enhance this point, discuss the role of specific nutrients and dietary patterns in influencing mood and anxiety in adolescents. Thank you for pointing out this. We have rectified this in our manuscript. See page no: 8

The discussion on tuition and academic stress is valuable. Expand on potential reasons why tuition attendance may be associated with higher anxiety levels. Discuss strategies to promote academic support while minimizing stress among students. Thank you for pointing out this. We have rectified this in our manuscript. See page no: 8

The comparison of BMI and anxiety levels is relevant. However, provide a more nuanced analysis of the relationship, considering potential confounding factors, such as age, gender, and lifestyle habits. Discuss how BMI changes over time may influence anxiety outcomes. This we discussed in little detailed in discussion part last paragraph at page # 9

Consider adding a final section summarizing the key findings and implications of the study. Discuss how the results contribute to the existing body of knowledge on adolescent anxiety and suggest potential interventions or areas for further research. 

Overall, the discussion is well-structured, and the inclusion of supporting studies adds credibility to the presented findings. To enhance the section, provide more in-depth analysis and interpretation of the results, while also highlighting potential avenues for future research and practical implications for mental health interventions. Thank you for highlighting this point. We have made changes in discussion part. Please check page no: 8

---

## [Editor Report · Decision Letter 3]

31 Jul 2023

Anxiety levels among school-going adolescents in peri-urban areas of Karachi, Pakistan

PONE-D-23-04889R3

Dear Dr. Sharif,

We’re pleased to inform you that your manuscript has been judged scientifically suitable for publication and will be formally accepted for publication once it meets all outstanding technical requirements.

Kind regards,

Sheikh Mohd Saleem, MBBS, MD

Academic Editor

PLOS ONE

Additional Editor Comments (optional):

Thanks for addressing the comments
---

## [Editor Report · Acceptance letter]

5 Oct 2023

PONE-D-23-04889R3 

Anxiety levels among school-going adolescents in peri-urban areas of Karachi, Pakistan 

Dear Dr. Sharif:

I'm pleased to inform you that your manuscript has been deemed suitable for publication in PLOS ONE. Congratulations! Your manuscript is now with our production department. 

Kind regards, 

on behalf of

Dr. Sheikh Mohd Saleem 

Academic Editor

PLOS ONE